# Effect of Estradiol on Estrogen Nuclear Receptors Genes Expression on Embryonic Development Stages in Chinese Soft-Shelled Turtle (*Pelodiscus sinensis*)

**Guobin Chen** [1,2]**, Tong Zhou** [1]**, Meng Chen** [1] **, Guiwei Zou** [1],*** and Hongwei Liang** [1],***

1    Yangtze River Fisheries Research Institute, Chinese Academy of Fisheries Science, Wuhan 430223, China
2    College of Fisheries and Life Science, Shanghai Ocean University, Shanghai 201306, China
*    Correspondence: zougw@yfi.ac.cn (G.Z.); lianghw@yfi.ac.cn (H.L.); Tel.: +86-27-81780097 (H.L.)

**Abstract:** Among Chinese soft-shelled turtles, *Pelodiscus sinensis*, males have a richer nutritional value and higher market price than females. All-male offspring were obtained by 17β-estradiol (E2). However, the molecular mechanisms of E2 inducing sexual reversal remain unclear. In this study, we cloned estrogen nuclear receptors (ERs) from *P. sinensis* and investigated their expression profiles. We examined the responses of ERα and ERβ after treatment with different concentrations of 1.0, 5.0, and 10 mg/mL E2. ERs showed abundant expressions in the adult gonad, *ERα* for ovary, and *ERβ* for testis. E2 can up-regulate the expression of *ERα*, which showed a remarkable increase while the promotion of *ERβ* was unobvious. They reached a high level at stage 17 after the treatment of E2, genes of the female-related genes *Rspo1*, *Wnt4*, β-catenin, *Foxl2*, *Cyp19a1*, and *Sox3* exhibited a significant raise at stage 17 with the increase in the concentration of E2 while the male-related genes *Sox9*, *Dmrt1*, and *Amh* were significantly inhibited. Our study cloned the full length of ERs and analyzed their structures and expressions, laying a foundation for the further study of the effect of estrogen on sex determination.

**Keywords:** Chinese soft-shelled turtle (*Pelodiscus sinensis*); sexual reversal; estrogen receptors; estrogen



## 1. Introduction

The Chinese soft-shelled turtle, *Pelodiscus sinensis*, is an economically important aquatic species that is widely distributed in China [1]. It has typical features of sexual growth dimorphism which mean males grow significantly faster and have richer nutritional value and higher market price than females [2]. In order to expand the breeding efficiency of breeding enterprises, the all-male breeding of *P. sinensis* using sex control approaches has become a practical requirement. Sex determination in reptiles can be divided into two categories. One is genotypic sex determination (GSD), in which sex determination genes on sex chromosomes first regulate and induce the cascade reaction of related sex determination genes, and finally regulate the development of primordial gonads towards the testis or ovary [3]. The other type is environment-dependent sex determination (ESD). The most typical ESD model is temperature-dependent sex determination (TSD), and the sex determination of embryos is affected by the environmental temperature during embryonic development, rather than genetic material. Researchers have confirmed that *P. sinensis* is a GSD species with ZZ/ZW heteromorphic micro-sex chromosomes [4], which was significantly different from the typical TSD in *Trachemys scripta* [5].

To obtain all-male offspring, E2 treatment is administered to male embryos (ZZ), E2 can induce gonads to develop into ovaries, but does not change the genotype, thus forming an individual with a female phenotype and male genotype called pseudo-female turtles (ΔZZ). These individuals could be used as the female parent while the male turtle was used as the male parent (ZZ) for breeding work, and their offspring would all be male [6,7]. In vertebrates, the sexual reversal process was co-regulated by the female-related genes

(*Rspo1*/*Wnt*/β-catenin/*Foxl2*/*Cyp19a1*/*Sox3*) and male-related genes (*Sox9*/*Amh*/*Dmrt1*), and the male-to-female sexual reversal could be caused after estrogen treatment during embryonic development [8]. *Rspo1*, known as a female-determining factor, functions upstream of the female sex determination pathway to activate the *Wnt*/β-catenin signaling pathway in mammals [9]. In humans, the deletion or mutation of *Rspo1* could lead to female-to-male sexual reversal [10]. In mice, the over-expression of β-catenin reversed the sex of male XY individuals to female XY individuals [11]. The *Sry*-related gene *Sox9* was involved in the differentiation of Sertoli cells in male gonads of vertebrates with different kinds of sex determination which could be inhibited by estrogen resulting in male-to-female sexual reversal [12]. The medaka, *Oryzias latipes*, the pattern of male-to-female sexual reversal could be achieved by estrogen treatment and its special sex determination gene named DM-domain gene on the Y chromosome (*Dmy*) was clearly suppressed during the embryonic development stage [13]. In Gulf pipefish, *Syngnathus scovelli*, males were exposed to estrogen, which led to the feminization of the male liver transcriptome in a pipefish having undergone sexual reversal [14].

The estrogen receptors expressed in vertebrates and a few invertebrates undertake the effect of estrogen on sex-related genes [15]. Estrogen receptors were mainly divided into two categories: classical estrogen nuclear receptors (ERs) and novel membrane receptors. The first estrogen binding protein discovered is known as estrogen receptor α (ERα, also known as ER1 or Esr1) [16]. The second nuclear estrogen receptor was named estrogen receptor β (ERβ, also known as ER2 or Esr2) [17]. Both of them are nuclear transcription factors that were involved in the regulation of many complex physiological processes [18,19]. In the nuclear, ERs act as ligand-dependent transcription factors, when the hormone signal enters the cytoplasm, it binds to the estrogen nuclear receptor, forming the hormone–nuclear receptor complex. It was transported through the nuclear envelope to the nucleus which regulated the transcription factor activity by recognizing specific DNA fragments, thereby achieving the transcriptional regulation of sex-related genes [20]. ERs also regulate gene expression via binding to other transcription factors such as nuclear transcription factor-κB (NF-κB) [21]. Researchers found that the other is novel membrane receptors, that estrogen could rapidly up-regulate endometrial cyclic adenosine monophosphate (cAMP) levels through the cell membrane binding site, and thus speculated the existence of the membrane estrogen receptor (mER) [22], which was later discovered as G protein-coupled estrogen receptor 30 (GPER30) [23]. By way of membrane receptors, *G-protein-coupled receptors* were activated after binding with ligands, and their coupled proteins linked hormones to cAMP in the cytoplasm, thus playing a regulatory role [8].

Exogenous estradiol (E2) is critical in the process of all-male offspring; however, the molecular mechanism between E2 and estrogen receptors in sex determination remains unclear. In the present study, we cloned the ERs from *P. sinensis* and characterized their expression in normal adults and embryos. We further investigated the effects of E2 on the expression of nuclear receptors in male embryos. The results will contribute to further investigations of sex determination, sex differentiation, and the breeding work in Chinese soft-shelled turtles.

## 2. Materials and Methods

### 2.1. Maintenance of the Chinese Soft-Shelled Turtles

One-year-old healthy turtles (3 males and 3 females, mean weight $1100 \pm 100$ g) and fertilized eggs were obtained by random sampling from Anhui Xijia Agricultural Development Company (Bengbu, Anhui, China). Following euthanasia by MS-222 (600 mg/kg), the turtles were dissected and eight tissues including the heart, liver, spleen, lung, kidney, brain, ovary, or testis were collected and stored in liquid nitrogen for RNA isolation. All animal handling and experimental procedures were conducted under the guidelines for the care and use of the laboratory and approved by the Yangtze River Fisheries Research Institute Animal Care Committee.

### 2.2. Estradiol Treatment

The fertilized eggs were incubated in the constant temperature humidity incubator (Xinmiao, Shanghai, China) and kept at 30 ± 0.5 °C and 80–85% humidity for 15 days. We diluted E2 (MedChemExpress, Wuhan, China) with ethanol (Xilong, Guangdong, China) into a reagent of 1 mg/mL, 5 mg/mL, and 10 mg/mL which were the experimental groups. We dipped a cotton swab in a small amount of hydrochloric acid (HCl) (Xilong, Guangdong, China), and gently smeared it on the soft-shelled turtle fertilized eggs to make the egg-shell soft at stage 12 (15 days) of embryo development which was the critical period of sex differentiation. A micro-syringe (Gaoge, Shanghai, China) was used to inject 5 μL of one of the E2 solutions into each fertilized egg, the control group was injected with an equal volume of ethanol, and the blank group does not do any processing [24]. Two hundred fertilized eggs were inoculated at each concentration, and the blank control group raised 200 fertilized eggs for sampling. The injected site was then disinfected with 50 mg/mL of ampicillin and sealed with paraffin, putting the eggs back in the constant temperature humidity incubator.

### 2.3. Samples Collection

Samples, i.e., the embryos, were collected from stages 13 to 20, the fertilized eggs were broken up with tweezers and the embryos were removed and placed in cryogenic tubes, first in liquid nitrogen, and at the end of sampling, all samples were placed at −80 °C. Twelve samples were collected at each stage. The sex of all embryos was detected using sex-related markers (Ps4085-F/R, COI-F/R) developed by our laboratory [2]. After sex determination, the number of male embryos at each stage varied from 3 to 8. In order to ensure the consistency of the number of samples at each stage, we randomly selected 3 male embryonic samples from each stage for further analysis. Male samples of the experimental group and control group were used to explore the expressions of ERs during the embryonic development stage after the E2 treatment of adult turtle samples was used to detect the tissue distributions of ERs. Blank group samples were used to detect the expression of ERs during the normal embryonic development stage. All adult turtles were targeted for RNA extraction from organs (i.e., heart, liver, spleen, lung, kidney, brain, ovary, or testis), while total embryos after sex determination were targeted for RNA extraction.

### 2.4. RNA Extraction

The samples were taken out from −80 °C and thawed on ice, and then the samples were quickly decomposed. The gonads of the embryonic samples and part of the tissue samples of the adult turtles were placed into a 2 mL EP tube with 1 mL Trizol kit (Life Technologies, Shanghai, China) and three grinding beads. Then, the tissue crushing instrument was used at 120 Hz for 60 s, the solution was absorbed into a new 1.5 mL EP tube, and then chloroform was added and shaken for 5 min and centrifuged for 10 min. The supernatant was absorbed into a new 1.5 mL EP tube after centrifugation and blended with an equal volume of pre-cooled isopropyl alcohol, and centrifuged for another 10 min. After the supernatant was aspirated, 1 mL of pre-cooled 75% ethanol was added to 1.5 mL EP for centrifuging for 5 min to wash the RNA precipitate. The washing procedure was repeated three times, and then the liquid was poured out and dried on the ultra-clean table, and sterile water was added to dissolve for 10 min. Then, the concentration of the extracted RNA was detected by a Nanophotometer NP60 spectrophotometer (Implen, Munich, Germany) and the integrity of the RNA was detected by 1% agarose gel electrophoresis.

### 2.5. Full-Length cDNA Cloning of ERs

The cDNAs were synthesized using Hiscript®III 1st Strand cDNA Synthesis Kit (+gDNA wiper) (Vazyme, Nanjing, China) according to the manufacturer's instructions. Sequences from conserved domain amplification, 3′ RACE, and 5′ RACE were assembled to generate the full-length cDNA. Then, polymerase chain reaction (PCR) amplification for the conserved domain was carried out in the following programs: 95 °C for 2 min; 35 cycles

of 95 °C for 30 s; 60 °C for 30 s and 72 °C for 30 s; and a final extension at 72 °C for 5 min. The PCR products were detected by 1.2% agarose gel electrophoresis and sequenced by the Tianyihuiyuan Biotech Company (Wuhan, China). The programs of 5′ and 3′ RACE were set as 94 °C for 5 min; followed by 5 cycles of 94 °C for 30 s and 72 °C for 3 min; 5 cycles of 94 °C for 30 s, 70 °C for 30 s, and 72 °C for 3 min; and 25 cycles of 94 °C for 30 s, 68 °C for 30 s, and 72 °C for 3 min. The PCR products were purified using a gel extraction kit (Omega, Wuhan, China) and the purified products were ligated to a pMD18T vector (Takara, Dalian, China), and the recombinant plasmids were subsequently transformed into competent Escherichia coli DH5α cells (Biomed, Beijing, China). Positive clones were screened and sequenced by the Tianyihuiyuan Biotech Company. Primers based on the transcriptome sequences (https://ngdc.cncb.ac.cn/gsa/, accessed on 10 November 2021 with accession number CRA005737) were designed and other primers for related genes were used (Table 1).

**Table 1.** Primers are used for PCR.

| Primer Name | Primer Sequence (5′–3′) | Application | Related References |
|---|---|---|---|
| *ERα*-F<br>*ERα*-R | GTTGATCCCTCCGCTGACAGT<br>CTCGCAAGACCAGACTCCATAAT | CDS amplification | |
| *ERβ*-F<br>*ERβ*-R | TGACGTTACTACAGCCAGCATCAC<br>CGACCTCCACATCAGACCCATC | | |
| *ERα*-GSP5-1<br>*ERα*-GSP5-2 | ATCTGGTGGAGCATGGCAACTC<br>GAATCTGGTGGAGCATGGCAAC | 5′ RACE | |
| *ERβ*-GSP5-1<br>*ERβ*-GSP5-2 | TAATCAAAGCTCGTGGAGTGGC<br>GCGTACGTGTATTTGTCGGTCA | | |
| *ERα*-GSP3-1<br>*ERα*-GSP3-2 | GCCAGTTAACAACTGCATCAACTT<br>AAGCAGGGAAGATGAGAATTTGC | 3′ RACE | |
| *ERβ*-GSP3-1<br>*ERβ*-GSP3-2 | ATGCTAGATGCTCACCGATTGC<br>CAGGCACATGAGCAATAAAGGG | | |
| UPM short<br>UPM long | CTAATACGACTCACTATAGGGC<br>CTAATACGACTCACTATAGGGCAAGCA | 5′ and 3′ RACE | |
| *ERα*-F<br>*ERα*-R | CCGACTGCGAAAGTGCTATGA<br>ACGCTGGACTGTTCTTCTTGCTA | | |
| *ERβ*-F<br>*ERβ*-R | GCAACAGACAACTCGCATGG<br>GTGTGTGCATTCAGCATCTCC | | |
| *Rspo1*-F<br>*Rspo1*-R | CCTGCTGGAGAGGAATGACA<br>CCCACTCGCTCATTTCACA | | [7] |
| *Wnt4*-F<br>*Wnt4*-R | GAGGTGATGGACTCGGTGCG<br>CCCGTTCTTGAGGTCGTGGTC | | |
| β-catenin-F<br>β-catenin-R | GCTTTGGGACTCCACCTTACAG<br>ATCACCAGCCCGAAGAACAGT | qPCR | |
| *Foxl2*-F<br>*Foxl2*-R | ATCTGTTTTTATTAGCACGGTT<br>CCTTCTCAGGAGGAGTTTCGT | | [25] |
| *Cyp19a1*-F<br>*Cyp19a1*-R | TCGTGGCTGTACAAGAAATACGAA<br>CCAGTCATATCTCCACGGCTCT | | [26] |
| *Sox3*-F<br>*Sox3*-R | GAGTGTAGAGGTGGAATGGAAACG<br>AAACCCTCAAGCAGGATACGG | | |
| *Sox9*-F<br>*Sox9*-R | TACGACTACACCGACCACCA<br>GTAGTGTCTGCAATGGGCGT | | |
| *Dmrt1*-F<br>*Dmrt1*-R | CCGCCTCGGGAAAGAAGTC<br>TGCTGGATGCCGTAGTTGC | | [27] |
| *Amh*-F<br>*Amh*-R | CGGCTACTCCTCCCACACG<br>CCTGGCTGGAGTATTTGACGG | | |
| *Ps4085*-F<br>*Ps4085*-R | GTTTGAAGTGCTGCTGGGAAG<br>TTCCCCGTATAAAGCCAGGG | Sex identification | [2] |
| *COI*-F<br>*COI*-R | CAACCAACCACAAAGACATTGGCAC<br>ACCTCAGGGTGTCCGAAAATCAAA | | |
| *Gapdh*-F<br>*Gapdh*-R | AGAACATCATTCCAGCATCCA<br>CTTCATCACCTTCTTAATGTCGTC | Internal control | |

*2.6. Sequence and Phylogenetic Analysis*

Sequences of ER protein from different species were downloaded from the NCBI database and aligned using DNAMAN. The protein domains were predicted using online software. A phylogenetic tree was then constructed by the neighbor-joining (NJ) method, with 1000 bootstrap replicates using MEGA 7.0 (http://www.megasoftware.net, accessed 13 December 2021).

*2.7. Gene Expression Analysis by Quantitative Real-Time Reverse Transcription-PCR*

The expression patterns of ERs genes and the transcriptional response to E2 were investigated based on quantitative real-time reverse transcription-polymerase chain reaction (qRT-PCR). Total RNAs were isolated from each tissue of the Chinese soft-shelled turtles according to the Trizol kit, 1500 ng/μL of total RNA was reverse-transcribed to single-strand cDNA using a HiScript®II Q RT SuperMix for qPCR (+gDNA wiper) (Vazyme) according to the manufacturer's instructions. The primers used for qRT-PCR (Table 1) were designed according to the ERs sequences. The *Gapdh* gene was used as an internal control. The qRT-PCR was performed using SYBR Select Master Mix (Vazyme) with the following thermal cycling conditions: 95 °C for 1 min, 40 cycles of denaturation at 95 °C for 15 s, annealing at 60 °C for 30 s. Each group was performed for three replicates to reduce the error of the experiment. The relative expression levels of ERs were normalized to that of *Gapdh* quantification by the $2^{-\Delta\Delta Ct}$ method. The same method was used to examine the expression levels of other genes.

*2.8. Statistical Analysis*

The qRT-PCR data were expressed as mean ± SD and statistical analysis (one-way ANOVA) was performed by one-way analysis of variance (ANOVA) followed by Duncan's multiple comparison test. The level of statistical analysis was set at $p < 0.05$ and was considered significant.

## 3. Results

*3.1. Sequence Analysis of P. sinensis ERs Gene*

The full-length cDNA sequence of *ERα* was 3014 bp, with an 900 bp 5′ untranslated region (UTR), a 260 bp 3′ UTR with poly (A) tail, and an open reading frame (ORF) of 1854 bp, coding 617-amino acid. ERα protein contained a DNA-binding domain (DBD; amino acids 203–284), a ligand-binding domain (LBD; amino acids 333–570), and a final C-terminal (amino acids 580–617). (Figure 1). The amino acid sequence of *P. sinensis* ERα shared 88.33, 82.80, and 79.77% identity with that of *Mauremys mutica*, *Chelonia mydas*, and *Chrysemys picta bellii*, respectively (Table 2). The phylogenetic analysis showed that *P. sinensis* formed a clade with other turtle species with high similarity (Figure 2).

**Table 2.** Comparative identity of the amino acid sequence of ERs.

| Species | ERα | | ERβ | |
| --- | --- | --- | --- | --- |
| | Accession Number | Identity (%) | Accession Number | Identity (%) |
| *Mauremys mutica* | XP 039386245.1 | 88.33 | XP 039390595.1 | 93.17 |
| *Chelonia mydas* | XP 007060399.2 | 82.80 | XP 007068065.2 | 86.35 |
| *Chrysemys picta bellii* | XP 042702245.1 | 79.77 | XP 005285947.1 | 87.54 |
| *Numida meleagris* | XP 021237930.1 | 76.01 | XP 021259551.1 | 80.03 |
| *Gallus* | NP 990514.1 | 74.28 | XP 040556512.1 | 80.38 |
| *Homo sapiens* | NP 000116.2 | 67.49 | NP 001428.1 | 69.11 |
| *Sus scrofa* | XP 020938661.1 | 67.49 | NP 001001533.1 | 70.31 |
| *Mus musculus* | NP 001289460.1 | 66.47 | NP 997590.1 | 72.01 |
| *Equus caballus* | XP 023488612.1 | 64.88 | NP 001296408.1 | 70.31 |
| *Danio rerio* | XP 009297713.1 | 41.18 | NP 851297.1 | 49.66 |

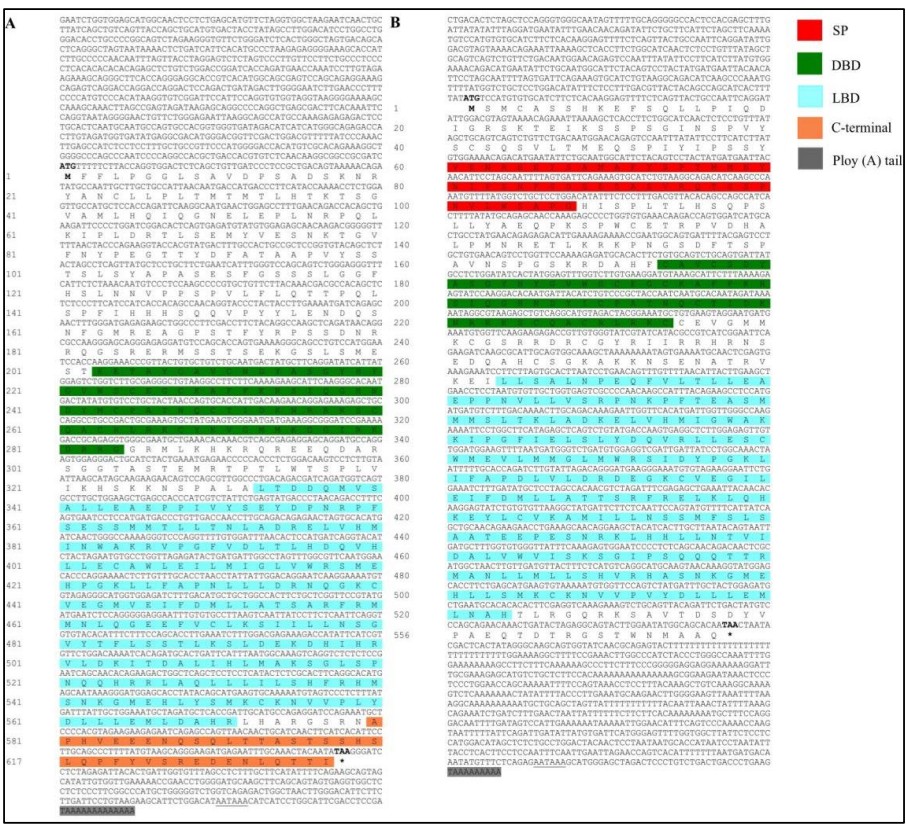

**Figure 1.** The cDNA sequence and amino acid of ERs of *P. sinensis*. The start codon and end codon are in bold black, * denotes the termination codon, and the underline denotes the 3′ untranslated region with the tailing signal AATAAA. SP: signal peptide. DBD: DNA-binding domain. LBD: ligand-binding domain. (**A**) The cDNA sequence and amino acid of *ERα*. (**B**) The cDNA sequence and amino acid of *ERβ*.

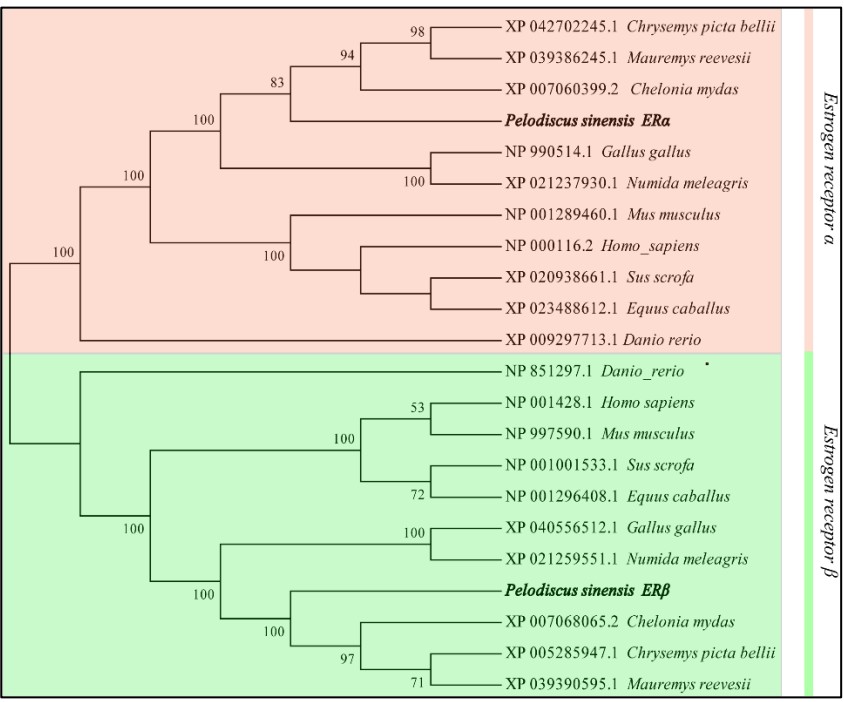

**Figure 2.** Gene tree of ER amino acid sequences.

Likewise, the full-length cDNA sequence of ERβ was 2950bp and comprised a 483-bp 5′ UTR, a 1671-bp open reading frame, and a 796-bp 3′ UTR. The 556-amino acid *ERβ* protein with an N-terminal (NH2) signal peptide (SP; amino acids 60–107), a DBD (amino acids 173–233), and an LBD (amino acids 305–523). The blast results show that the ERβ protein sequence had a high identity with, *M. mutica*, *C. bellii,* and *C. mydas* of 93.17, 87.54, and 86.35%, respectively (Table 2).

### 3.2. Tissue and Embryonic Development Stage Distribution of ERs

The expression of ERs in different tissues and embryonic development stages between ZZ-male and ZW-female were evaluated by qRT-PCR. In normal adults, ERs were detected in the heart, liver, spleen, lung, kidney, brain, muscle, and gonad. Both *ERα* and *ERβ* had the highest expression in the gonad while the muscle had the lowest level for *ERα*, the heart for *ERβ* in females (Figure 3A), the lung for *ERα,* and the spleen for *ERβ* in males. *ERβ* had a higher expression value than *ERα* only in the testis and kidney (Figure 3B). The ERs expression of embryonic development stages between ZZ-male and ZW-female in the blank group held a prominent discrepancy that *ERα* had higher expression levels than *ERβ* across all developmental stages for both sexes in the blank group, suggesting that *ERα* may play a greater role during these developmental stages (Figure 3C,D).

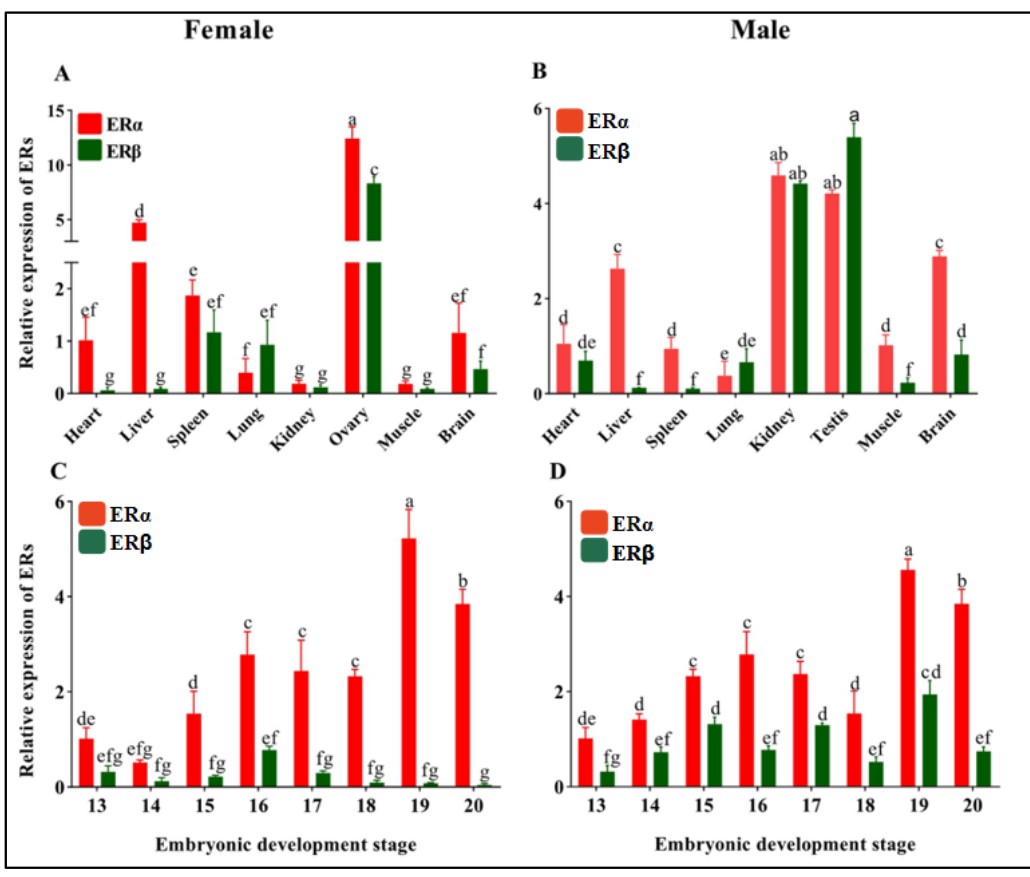

**Figure 3.** Expression profiles of ERs cloned from *P. sinensis* in normal adult females (**A**) and male tissues (**B**), and expression analysis in blank group female embryos (**C**) and male embryos (**D**) at different embryonic development stages. Data are mean ± SD (n = 3). Significant differences at *p* < 0.05 are labeled with different letters.

### 3.3. Effect of Estradiol on ERs Expression and Sex-Related Genes

To analyze the influence of E2 on ERs, their expressions were investigated in the experimental group during embryonic development in the Chinese soft-shelled turtles throughout qRT-PCR. The relative expressions of ERs after treatment with different concentrations of E2 were measured (Figure 4). Compared with the control group, the expression of *ERα* gradually increased from stage 13 to 20, which was more obvious after stage 14. The expression of *ERα* was markedly up-regulated and peaked at stage 17 after the injection of 1 mg/mL and 5 mg/mL E2 (Figure 4A,B) while the expression of *ERβ* owed a stable increase (Figure 4D,E). The expression of *ERα* was enhanced and peaked at stage 19 after treatment with 10 mg/mL E2, and *ERβ* expression was significantly increased and peaked at stage 19 during the embryonic development stage (Figure 4C,F). Additionally, the expression level of *ERα* peaked at stage 17 with the concentration of E2 for 1 mg/mL, and 5 mg/mL and it would continue to grow to stage 19 for 10 mg/mL. Meanwhile, *ERβ* also reached its highest expression level at stage 17 for 10 mg/mL. Thus, to further explore the effect of ERs on sex differentiation, sex-related genes that responded to E2 were measured at stage 17 (Figure 5). The expression levels of *Rspo1*, *Wnt4*, β-catenin, *Foxl2*, *Cyp19a1*, and *Sox3* had an increased expression with increasing concentrations while *Sox9*, *Dmrt1*, and *Amh* were significantly inhibited.

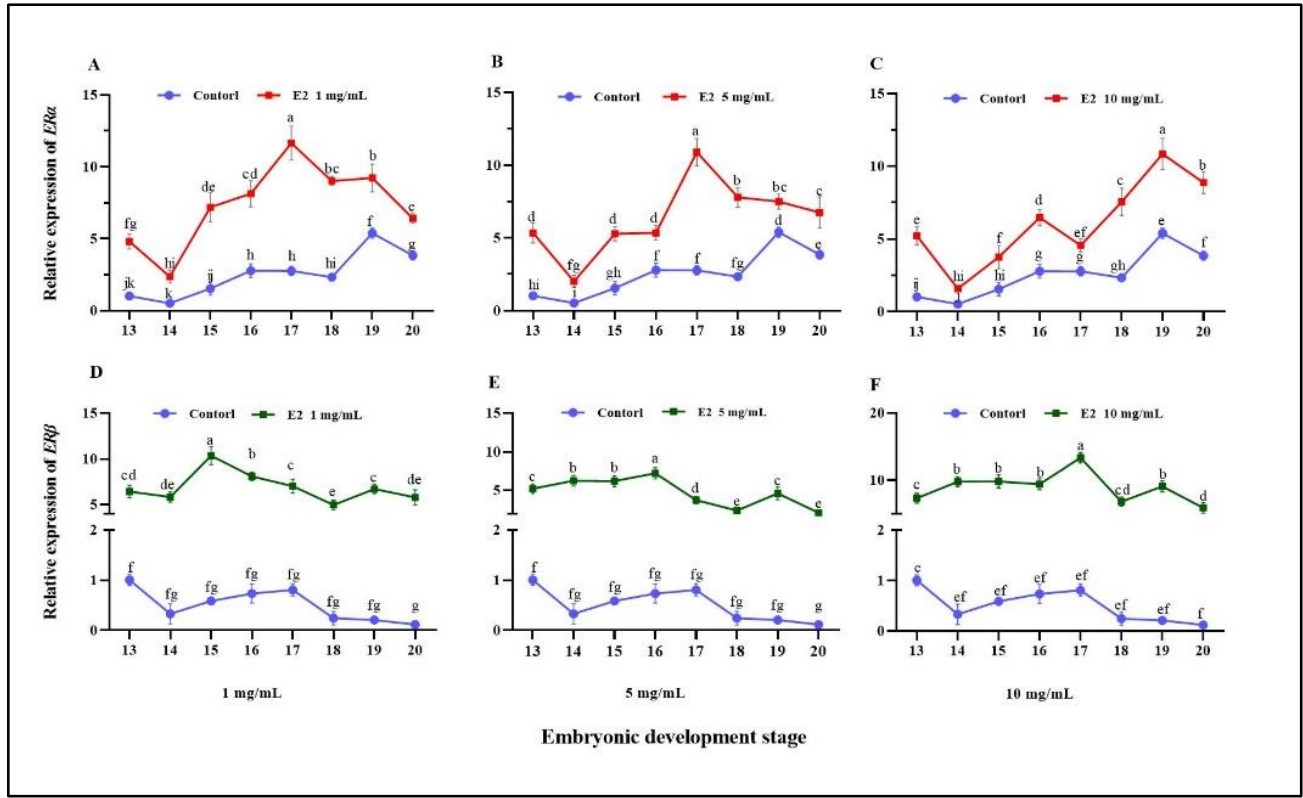

**Figure 4.** Expression of ERs in E2-treated male embryos at embryonic development stages after injection with different concentrations of E2. Data are the mean ± SD (n = 3). Different letters indicate significant differences between the control group and the experimental group, respectively ($p < 0.05$): (**A**) *ERα* for 1 mg/mL E2; (**B**) *ERα* for 5 mg/mL E2; (**C**) *ERα* for 10 mg/mL E2; (**D**) *ERβ* for 1 mg/mL E2; (**E**) *ERβ* for 5 mg/mL E2; and (**F**) *ERβ* for 10 mg/mL E2.

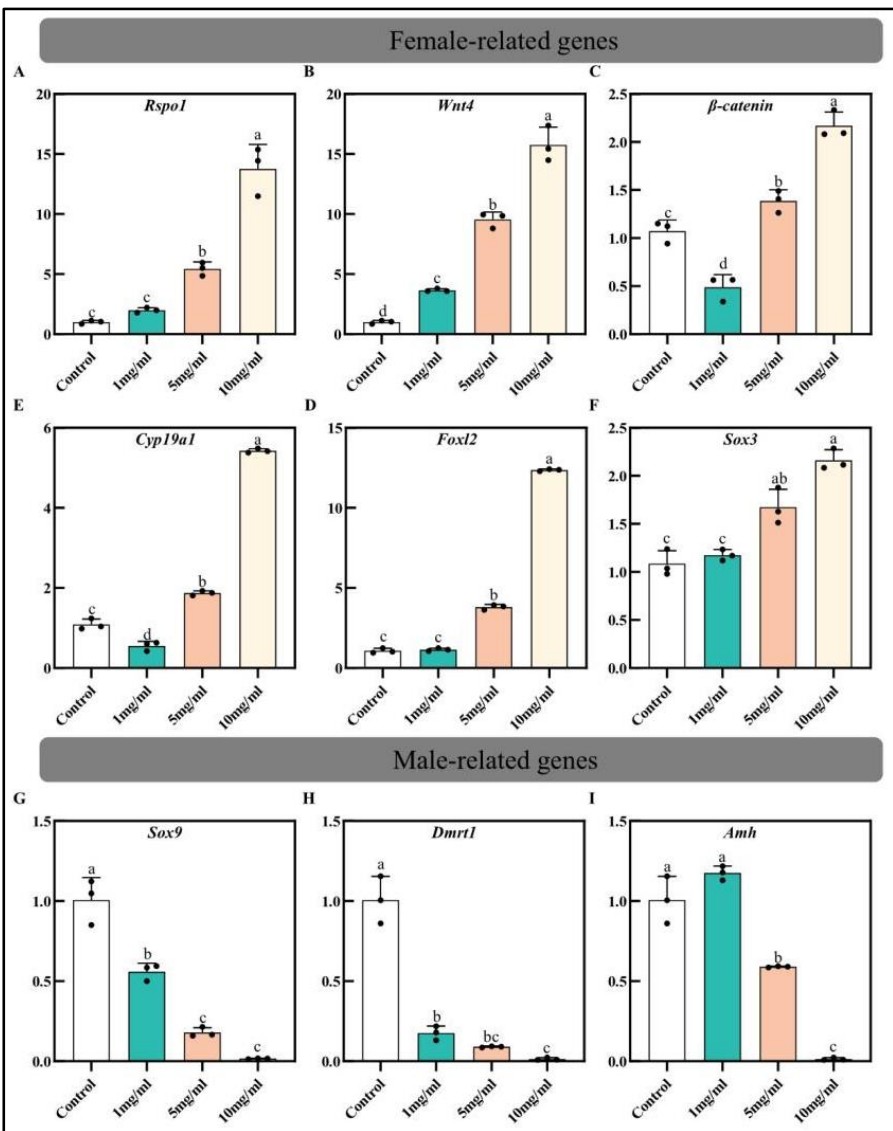

**Figure 5.** The expressions of sex-related genes at stage 17. Female-related genes: (**A**) *Rspo1*; (**B**) *Wnt4*; (**C**) β-catenin; (**D**) *Foxl2;* (**E**) *Cyp19a1*; and (**F**) *Sox3*. Male-related genes: (**G**) *Sox9*. (**H**) *Dmrt1* (**I**) *Amh*. Data are the mean ± SD (n = 3). Significant differences at *p* < 0.05 are labeled with different letters.

## 4. Discussion

Sex determination and the gonadal development mechanism in *P. sinensis* were of concern because of the economic characteristics associated with significant sexual dimorphism [28]. To obtain all-male offspring, the pseudo-females (ΔZZ) could be obtained by the treatment of E2 to male embryos, then the pseudo-females (ΔZZ) and male (ZZ) can produce all-male individuals [6,24]. In the present study, the full-length cDNA sequences of *ERα* and *ERβ* were cloned and their expressions in normal adult tissues and embryonic development stages were investigated.

RT-qPCR analysis in this study revealed that ERs exhibited different expression profiles in the tissues of females and males, indicating their different physiological functions [29]. Both of them were detected in the gonads with high expression levels, suggesting that they were related to the development of gonads and sex differentiation. Even knocking down *estrogen receptor-related factor* (ERRF) in *Drosophila* led to improperly developed testis [30]. In the present study, the expression of *ERα* was higher than *ERβ* in the ovary, suggesting that *ERα* may play a critical role in early female sex differentiation and ovary development. There have been articles that suggest *ERα* plays a major role in the differentiation of the

gonad of *P. sinensis* [31]. *ERβ* was mostly distributed in the ovary and testis which was similar to *Mandarin fish* [32]. In *P. sinensis*, it has been shown that *ERα* agonists could induce ovarian differentiation and sexual reversal [33]. In mammals, the expression of *ERα* was located in the interstitial cells and germinal epithelium, and *ERβ* was expressed in mesenchymal–epithelial cells of the early ovarian development phase, indicating that both of those were indispensable for germ cell differentiation and ovarian lumen development [34,35]. The expression level of *ERβ* in the testis was higher than that of *ERα* and showed a high level in the kidney. Previous research showed that the expression level of *ERβ* represented obvious sex differences which were 4–8 times higher in males than in females [36]. In teleost fishes, there is another type of ER named *ER-γ* whose DBD (ERβ for amino acids 155–234, ERβ2 for amino acids 171–252) shared 88.4% identity and had similar expression levels in the pituitary [20] and has been re-named *ERβ2* [37]. The similarity between *ERβ* and *ERβ2* seems to be specific for teleost fishes likely due to a gene replication event causing the receptor's emergence. In tilapia, the homozygous mutants of *ERβ* resulted in decreasing spermatogonia and an abnormal increase in spermatozoa, and the mutation of *ERβ2* could cause reproductive tract malformations, indicating that *ERβ* was critical for spermatogenesis while *ERβ2* was indispensable for the development of a male reproductive organ [38]. Additionally, the high expressions of *ERα* and *ERβ* were observed in the spleen, brain, and liver which were similar to the results in mice and humans [39,40].

Estrogen exerted a variety of important physiological effects, which have been suggested to be mediated via the two known ERs [41]. In this study, *ERα* played a major role during the embryonic development stages of both sexes and had a more marked change than *ERβ* after the stimulation of E2, which suggested that *ERα* and *ERβ* may play different roles and even have antagonistic effects. In American alligators, studies have shown that it was *ERα*, rather than *ERβ*, that regulated the sexual reversal induced by estrogen [42]. In mice, *Erα*-mediated estrogen effects in female reproductive tracts while *ERβ* antagonized *ERα*-mediated estrogenic action [43]. Moreover, *ERβ* inhibited *ERα*-mediated gene transcription in the presence of ERα, whereas, in the absence of ERα, it could partially replace *ERα* and even regulate the transcriptional activity of *ERα* [44]. In *Branchiostoma belcheri*, the staining of the related cells of gonads revealed that the *ERα* and *ERβ* co-existed in the same cell, but the target cell localization was different, suggesting that *ERα* and *ERβ* may have different roles in the mediated estrogen signaling pathway and the mechanism of gene transcription [45]. In tilapia, mutations in *ERα* caused a loss of reproductive function in both males and females, while *ERβ* mutants showed significantly delayed ovarian development and follicle growth in females while males showed fewer spermatogonia and more abnormal sperms. The *ERβ2* mutants displayed the abnormal development of the ovary and testis, resulting in infertility in females and males, respectively, although they produced gametes in their gonads [38]. These mutation results of ER subtypes further suggest that different ERs may play different roles during embryonic development stages [46].

Furthermore, the sex-related genes were detected at stages 17 after the E2 treatment to male embryos, but the expression of genes showed a different expression change with ERs. With the increase in E2 concentration, the expression levels of female-related genes *Rspo1/Wnt4/β*-catenin/*Foxl2/Cyp19a1* were clearly increased. Conversely, the expression levels of male pathway genes such as *Sox9/Dmrt1/Amh* were significantly inhibited. Compared with ERs, the expression levels of genes of male and female pathways changed differently after E2 treatment. At stage 17, with the increase in E2 concentration, the expression level of *ERα* compared with *ERα* under 1 mg/mL and 5 mg/mL E2 treatment was decreased, this is different from the higher expression of female sex-related genes and the lower expression of male sex-related genes with increasing concentration. Moreover, the peak value stage of ERs clearly shifted backward, which was *ERα* for stage 17, stage 17, and stage 19 while *ERβ* for stage 15, stage 16, and stage 17 under 1 mg/mL, 5 mg/mL and 10 mg/mL E2, respectively. This may be because sex-related genes were not only regulated by ERs but also undertook the effect of some other factors such as nuclear transcription

factor-κB (NF-κB) [21,47] which was generally referred to as "transcriptional cross-talk" [48]. When the ERs bond to the NF-κB, inhibition of NF-kB activity by *ERα* and *ERβ* would be caused if the AF-1 of ER was activated [49,50]. The ability of *ERα* to activate cooperatively transcription with NF-kB required the AF-1 domain [51]. Thus, due to the combination of NF-kB and ERs, the signal entering the nucleus to bind to ERs would be reduced, resulting in different expression trends of ERs and sex-related genes.

The sex determination and sex differentiation of animals are very complex physiological processes, focusing on the mechanisms underlying sex differentiation by the ERs will aid in the development of effective breeding strategies. *ERα* and *ERβ* are necessary for mediating the effects of E2; in the present study, *ERα* may play a more important role than *ERβ* during embryonic development stages. Their detailed molecular mechanisms in the sexual reversal process and reproduction deserve further study. Subsequent studies can specifically explore their functions by knocking down individual nuclear receptors and the experimental research results will hopefully serve as useful feedback information for improvements for the Chinese soft-shelled turtle's gonadal differentiation and breeding efforts.

## 5. Conclusions

In this study, the pseudo-females could be obtained by injecting estradiol, in which the sex was reversed with the male genotype and female phenotype. The full-length of *ERα* was 3014 bp, while *ERβ* was 2095 bp. After treatment with different concentrations of E2, the expression levels of *ERα* and *ERβ* were enhanced clearly. Besides, female sex-related genes *Rspo1*/*Wnt4*/β-catenin/*Foxl2*/*Cyp19a1* were also increased while male sex-related genes *Sox9*/*Dmrt1*/*Amh* were down-regulated significantly. This study provides a reference for further investigations of the molecular mechanism of sex determination and all-male breeding of *P. sinensis*.

**Author Contributions:** G.C.: operated the experiments, data curation, and writing—original draft preparation. T.Z.: assisted sample collection and revised the manuscript. M.C.: performed the PCR experiments. G.Z.: provided the experimental consumables. H.L.: devised the experiment and revised the manuscript. All authors have read and agreed to the published version of the manuscript.

**Funding:** This work was supported by the Central Public-Interest Scientific Institution Basal Research Fund; CAFS (No. YFI202212 and No. 2020TD33); and the National Freshwater Aquatic Germplasm Resource Center (FGRC18537).

**Institutional Review Board Statement:** This study was conducted according to the appropriate Animal Experimental Ethical Inspection of Laboratory Animal Centre of the Yangtze River Fisheries Research Institute, Chinese Academy of Fishery Sciences (Wuhan, China) (ID Number: 2022YFI-ZT-01).

**Conflicts of Interest:** The authors report no declarations of interest.

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
