# Peer review of "Effect of Estradiol on Estrogen Nuclear Receptors Genes Expression on Embryonic Development Stages in Chinese Soft-Shelled Turtle (Pelodiscus sinensis)"

_fishes, doi:10.3390/fishes7050223_

Round 1

Reviewer 1 Report

Major comments

1. line 12: All-male offspring? I believe "All-female offspring" is correct

2. throughout the text: gene name should be italic and protein name should not be italic

3. lines 242-243: Estrogen receptorr in Drosophila ? are you sure?

4. line 254: any clear evidence of ER-g function?

5. recently following 2 closely related manuscripts have been published, therefore, the authors need adequately cite and discuss their and your own data .

Wang et al. (2022) Molecular characterization and functional analysis of Esr1 and Esr2 in gonads of Chinese soft-shelled turtle (Pelodiscus sinensis). J. Steroid. Biochem. Mol. Biol., 222, 106147.

Li et al. (2022). Esr1 mediates estrogen-induced feminization of genetic male Chinese soft-shelled turtle, Biol. Reprod.

Minor comments

1. lines 15 and 16: estrogen nuclear receptors a should be ERa and ERb, respectively

2. line 18: markable should be remarkable

3. line 30 and throughout the text: add a space before reference, [1]

4.  line 66: estrogen receptor a (ERa) should be estrogen receptor a (ERa)

5. line 67: estrogen receptor b (ERb) should be ERb

6. line 70: can not understand the sentence: estrogen-formed hormone-nuclear receptor complexes  need to revise the sentence

7. line 78: add estrogen before receptor

8. line 82: add (E2) after estradiol

9. line 83: estradiol should be E2

10. line 85: nuclear estrogen receptors should be ERs

11. line 92: add (city, China) after Company

12. line 93 and throughout the text and figure legends: add a space after value, 600

13. line 100: delete C after 30

14. line 100 add (company name, city, state, country)

15. line 114 and throughout the text: delete a space between value and %

16. line 120: add a space after value throughout the text, 30s should be 30

17. line 128: Escherichia coli should be italic

18. line 150: delete ", Nanjing, China"

19. line 153: Gapdh should be italic

20. line 159: Result should be Results

21. line 160: P. sinensis should be P. sinensis

22. line 165: P. sinensis should be P. sinensis

23. line 166: should be 92.04, 91.88, 91.24, 86.78, and 83.92%

24. line 166: Chelonia mydas should be italic

25. line 167  should be ...  elegans, and Alligator sinensis, respectively

26. line 168: ERa should be ERa

27. line 169: Chelonia mydas should be C. mydas

28. line 175: ERb should be ERb

29. line 176: Trachemys, Mauremys, Chelonia should be T. M. C.

30. line 177 should be 92.46, 92.10, and 90,66%

31. line 179: P. sinensis should be italic

32. line 216: estrogen receptors should be ERs

33. line 242: Drosophia   ERs should be italic

34. line 255: estrogen receptor should be ER

35. line 265: nuclear estrogen receptors should be ERs

36. line 283: estrogen receptor should be ER

37. line 284: estrogen receptors should be ERs

38. line 295: estrogen receptor should be ER

39. References: the authors need to follow journal style for preparation of references: year of publication should be bold, volume should not be bold, journal names should be appropriately abbreviated, need to add pages, titles should not be large capital

Reviewer 2 Report

The authors performed a study into the effects of exogenous estradiol on estrogen receptor expression in soft-shelled turtles. The authors noted the expression patterns of the two paralogs of ER genes (alpha and beta). They started by targeting the sequence of ER in the species and running a gene tree analysis to confirm the placement of the ERs. The data they present show slight differences in ERalpha and ERbeta based on sex, tissue, and development stage. The patterns they present are mostly in line with the literature that exists. They further tested genes associated with sex differentiation and found E2 increases female-biased genes and also decreases male-biased genes.

I feel this paper is not ready for publication. I am unclear on the motivation for the study other than these expression patterns have not been reported for P. sinensis. The writing is uneven and hard to follow. This makes understanding the purpose challenging. The basic fact patterns are in the introduction and discussion, but more needs to be done to smooth out the logic and goal of the authors.

Some examples of odd phrasing include:

“Different pattern with ER”, what does this mean? Sex-biased genes expression patterns do what would be expected with E2 exposure—female up-regulated and male down-regulated

“It has been proved”, it might be better to say “a study showed” or “this pattern exists in” as proved is a loaded term.

“Furtherly” is not a word, there were other types of expressions that did not fit.

It is especially frustrating when reading the methods to understand what the authors have done. The authors do an insufficient job of describing sampling, for example no description on number of eggs, which genetic sex, where they were obtained, or how they relate to the adult samples. Further, were the same eggs sampled multiple times or sacrificed for collection? Figure 3 is another example of lack of clarity: “expression analysis in adult females at different developmental stages.” Were the adults exposed to E2? Where is the data on E2 exposure on ER expression in genetic females (was it even done?) There are no methods listed for the series of gene expressions the authors tested.

In the gene tree the authors make they should be clear that this is a gene tree, not a species tree. If the authors want to confirm the match to a species tree with current knowledge, I would bet there is a published species tree available. P sinensis in not more closely related to C. mydas than any other turtle species. The most recent common ancestor P. sinensis has with C. mydas is shared with ALL turtle species (again, this is a gene tree)

All of these issues pile up to a paper that cannot be verified. The authors should go back and examine what their goals are and write the paper that establishes and discusses them. Importantly, the authors must clean up the methods and results so that there are no misunderstandings, missing methods, or even potential missing data (the last one I can’t be sure of). It is my belief the authors have done good work and have data worthy of publication, but need to package the manuscript in a more transparent and comprehensive fashion. Until they do, this paper cannot be published in this form

Round 2

Reviewer 2 Report

I thank the authors for addressing some of my concerns on this paper. The manuscript is now in a better place to be understood by readers, but there are still a few issues that need to be addressed by the authors. Again, the methods leave me with some questions (though not nearly as many as before). Below I list some the concerns and following those, some general comments to improve the readability of the manuscript. I think this is much closer to publication. I hope my suggestions and edits are of use to the editors and authors.

Sampling: The authors have gone to great lengths to clarify their sampling process, and I appreciate the effort. The embryo sampling is still not fully clear though. There is no section in the methods declaring how RNA was collected from the embryos. The authors provided feedback in their response, but that did not make it into the manuscript. The authors must provide these details in the manuscript. The authors again did not answer where the RNA was specifically collected from. Was the whole embryo ground up? Was fluid extracted? Was an organ targeted? These matter, especially given the authors’ demonstration of tissue-specific expression patterns in adults. The authors also state 12 samples were collected at each stage and only males were used using sex-markers. How many males were used at each stage? Looking at Figure 4 I would guess 3 males were used. Why this number? The odds of only 3 males in 12 samples are low (<10%). Did authors preselect the number 3? Did they choose 3 males to measure expression? Did they measure more than 3 males for that time point then only report 3? The first two scenarios are not a problem, but the last one is, therefore the authors need to state exactly how and why they choose the number of males for each time point.

Blast and Phylogeny:

Why did the numbers change between manuscripts in Table 2? The authors state they reanalyzed, but my objection wasn’t the numbers it was the authors’ misinterpretation. Now I’m wondering how the numbers changed so drastically between this and the previous version. This doesn’t have to be answered in the manuscript, but should be accounted for. My objection in the phylogeny is the authors’ mis-statement of how to read one, an error they commit again here. The closest relative to any species/gene is the one which shares the most recent node with the focal species/gene. Look at Figure 2. Follow the line coming out from P. sinensis until it reaches the first node (for ER1 that is the one with Bootstrap support 83). Everything to the right of that node is the closest relative. In this case that is three species. ALL of those species are P. sinensis closest relatives. The species with the highest similarity in BLAST is not the closest relative, it’s just the one with the closest match. The phylogeny tells you who the relative is, and in this case it is ALL THREE SPECIES OF TURTLE. The authors can just drop the sentence at Line 180 (“The phylogenetic tree…”) and the reported results are fine.

Line 30: Change “that males grow” to “in which males grow”

Line 31: Remove “Therefore,” and start with “In order to…”

Line 33: Remove “The way of” and start with “Sex determination…”

Line 44: Change “after the E2 treatment” to “an E2 treatment is administered”

Line 44: Need a comma after “embryos (ZZ)”

Line 45: Change “in the direction of the ovary” to “into ovaries”

Line 47: Change “turtles (deltaZZ) and could” to “turtles (deltaZZ). These individuals could”

Line 62: The species is “Gulf pipefish, Syngnathus scovelli”

Line 63: Change “estrogen leading to” to “estrogen, which led to”

Line 64: I don’t know what the authors are saying here. Rework this sentence

Line 66: Change “two categories. One was classical estrogen nuclear receptors (ERs).” to “two categories: classical estrogen nuclear receptors (ERs) and novel membrane receptors.”

Line 67: Change “protein was discovered and known” to “protein discovered is known”

Line 68: Change to “estrogen receptor discovered was named”

Line 70: Change “were” to “are”

Line 70-74. Nuclear receptors bind their ligands in the cytoplasm and shuttle them through the nuclear envelop into the nucleus where they bind to the genome. They do not bind their ligands in the nucleus

Line 75: Change “Besides, ERs also regulated gene” to “ERs also regulate gene”

Line 76: Start sentence with “Researchers found…”

Line 80: Change “In the way of” to “By way of”

Line 84: Change “estradiol (E2) caught a critical position in the” to “estradiol (E2) is critical in the”

Line 85: Change “offspring. However,” to “offspring; however,”

Line 87: Change “Furthermore, we investigated” to “We further investigated”

Line 94: Change “the fertilized eggs were obtained randomly from” to “fertilized eggs were obtained by random sampling from”

Line 95: This sentence can be removed

Line 96: Change “After anesthesia” to “Following euthenasia”

Line 106: Change “experimental group then dipped” to “experimental groups. We dipped”

Line 111: Change “uL of E2 into” to “uL of one of the E2 solutions into”

Line 112: Here’s where the authors need to put their explanation of egg-breaking, preservation, and extraction.

Line 117: Remove “In this study” and start with “The sex…”

Line 118: Here author can discuss male numbers

Line 124: Again which tissues in embryo

Line 164: Does this mean each sampling was done three times from one tissue to confirm values or three different individuals were sampled once?

Line 166: Here the authors can state they used the same method for all other genes they tested

Line 197: Change “Figure 2: Phylogenetic trees of ERs amino acid sequence” to “Figure 2: Gene tree of ER amino acid sequences”

Line 204: Change “Besides ERB caught a stronger expression signal than” to “ERB had a higher expression value than”

Line 205: Change to “ERalpha had higher expression levels than ERbeta across all developmental stages for both sexes in the blank group, suggesting ERalpha may play a greater role during these developmental stages.”

Line 218: Change “were detected” to “were measured”

Line 228: Change “to furtherly explore” to “to further explore”

Line 229: Change “were detected” to “were measured”

Line 230: Change “were enhanced” to “had increased expression”

Line 232: Need to label x-axes

Line 243: Change “were widely concerned because” to “were of concern because”

Line 257: Change “articles to prove that ERalpha” to “articles that suggest ERalpha”

Line 259: Change “has been proved that” to “has been shown that”

Line 263: Change “both of them were” to “both of these were”

Line 264: Drop “Moreover,” and start with “The expression…”

Line 281: Drop “different concentrations of”

Line 283: Change “ERalpha rather than ERbeta regulated” to “ERalpha, rather than ERbeta, that regulated”

Line 291: Drop “Moreover,” and start with “In tilapia…”

Line 292: Change “and females. The ERbeta mutant females showed significantly delayed ovarian development and follicle growth while males” to “and females, while ERbeta mutants showed significantly delayed ovarian development and follicle growth in females while males”

Line 294: Drop “In contrast, the” and start with “ERbeta2…”

Line 297: Change “The mutation results of ER subtypes further proved that” to “These mutation results of ER subtypes further suggest that”

Lines 299-312: There are some typographical issues here, but my main concern is comparing sex-regulating gene response and ER response in the presence of E2. The authors say they are different, and I’m not entirely on board with this framing. ER clearly increases with E2 as does female-regulating. Are the authors suggesting that each E2 dose leads to a subsequent increase in female genes whereas in ER that doesn’t happen, (i.e. at Stage 17 the 10mg/mL dose yields the lowest ER expression). While that is true, it is clear that adding E2 increases ER compared to control. The authors need to be clear here. It sounds as if they are suggesting E2 doesn’t influence ER when it very much does. I would bet some sampling issue drove this discrepancy. The second half of the paragraph discussing alternate pathways is fine.
